# Lessons Learnt during COVID-19 Lockdown: A Qualitative Study of South African Families

**DOI:** 10.3390/ijerph182312552

**Published:** 2021-11-29

**Authors:** Gift T. Donga, Nicolette V. Roman, Babatope O. Adebiyi, Bernard Omukunyi, Rachel Chinyakata

**Affiliations:** The Centre for Interdisciplinary Studies of Children, Families and Society, University of the Western Cape, Robert Sobukwe Road, Bellville 7535, South Africa; nroman@uwc.ac.za (N.V.R.); atommega@yahoo.com (B.O.A.); bomukunyi@uwc.ac.za (B.O.); chin.rachie@gmail.com (R.C.)

**Keywords:** families, lessons learnt, COVID-19, lockdown

## Abstract

In a pandemic, such as COVID-19, with every single person struggling to deal with the unknown, it is often within the family that support is found but it is also within the family that circumstances, contexts and behaviours could further drive the pandemic and where they struggle to cope. This is novel research in the South African context with no known information regarding family life during and post the pandemic. This study, therefore, explores the lessons learnt during COVID-19 by South African families. A qualitative approach was employed to guide the gathering and analysis of the data. Data were collected from a sample of 31 family members above the age of 18 years from communities of the Western Cape Province and analysed through thematic analysis. According to the participants interviewed some of the significant lessons learnt from the lockdown include hygiene and health consciousness, appreciation for family, valuing life, self-introspection, less dependency, remote working, and financial savings. The realisation of such lessons even post-pandemic has the potential of strengthening families to be a resource of coping and resilience during very difficult times at the same time, contributing to greater physical, social, and economic functioning of families across South Africa.

## 1. Introduction

The coronavirus (COVID-19) pandemic is of a magnitude most individuals today have never witnessed. The outbreak was publicly reported to arise in late 2019, on 31 December 2019, a case of pneumonia with clinical symptoms of patients such as dyspnoea, dry cough, and fever whose unidentified cause was discovered in Wuhan City, Hubei Province in China [1]. COVID-19 was later declared a worldwide pandemic in March 2020 [2]. Up to date, a total of about 250 million cases of COVID-19 have been confirmed globally and more than 5 million people have succumbed to the virus [3]. Initially, there was widespread global concern regarding the virus reaching resource-constrained countries such as African countries, including South Africa. The reasons for these concerns are based on these countries having multiple health challenges such as larger prevalence rates of HIV/AIDS, tuberculosis, weak health systems and the increased movement of people. However, at the time of writing, the third wave had hit South Africa badly as infections kept on rising daily. By late September 2021, almost three million confirmed COVID-19 cases and more than 85,000 deaths were reported in the country [3]. During the same period, on average the daily new positive COVID-19 cases stood at 3300 and out of the country’s nine provinces, the Western Cape province contributed to the national figure an average of about 500 daily new cases [4]. While these statistics may seem better in comparison to other countries globally, the effects for people in South Africa have devastating consequences.

In this period of crisis, the world is often reminded of Winston Churchill’s famed quote during the second world war: “If you’re going through hell, keep going” [5]. Whereas South Africa is not facing a physical war, it is fighting the COVID-19 calamity in full force and like other nations, the country could not escape the pandemic as it suffered the loss of lives and livelihoods. To curb the sporadic spread of the virus, at one point most countries introduced control measures like the hard lockdown, travel restrictions, border closures and extensive prohibitions on nonessential commercial activities and social gatherings. South Africa is no exception and has also imposed extraordinary measures to control the spread, including the suspension of religious gatherings. The measures were carefully implemented through the introduction of the five-level COVID-19 alert system (see Figure 1) to control the progressive lockdown easing. These risk-adjusted criteria were informed by several factors, including the degree of infections and transmission rate, health facilities’ capacity, the extent of the enforcement of public health mediation and the social and economic effects of persistent restrictions [6].

On 27 March 2021, SA went into the first hard lockdown (alert level 5), which was later lowered down gradually as the cases of infections and rate of transmission went down until it reached level 1 in September 2020 [7]. In December 2020, the country experienced the second wave of COVID-19 infections leading to the retightening of restrictions to alert level three, then as the rate of new cases reduced, the lockdown restrictions were lowered to adjusted level 1 in March 2021 [8]. Due to the fast-spreading Delta variant that was first recorded in March 2021, the country moved from level 1 to an adjusted level 2, 3 and 4, respectively, during the peak of the third wave between May and July 2021 [9]. However, as the cases decreased and the vaccination roll-out was picking pace, by end of October 2021 the lockdown restrictions had been eased to adjusted level 1. Though laudable, the reality at play is that such unparalleled containment measures as those imposed on alert levels 4 and 5 reduced access to social and economic support systems for most South African families and communities at large. With the introduction of hard lockdowns and long curfews to contain the spread of the virus, families, particularly those staying in low-income communities were greatly impacted in the way they earn a living and were subjected to anxiety and depression [10]. Financial concerns, quarantine, and isolation are some main contributing factors for psychological and emotional distress and vulnerability among family members during this pandemic [11,12].

The family, consisting of self-defined members can be described as: “two or more individuals who depend on one another for emotional, physical, and economic support” [13] (p. 1). While family structures in South Africa are diverse in nature, many individuals depend on families and households for their physical, social, and economic functioning and regard families and households as their primary social institution [14]. In this COVID-19 pandemic, with every single person struggling to deal with the unknown, it is often within the family that support is found but it is also within the family that circumstances, contexts and behaviours could further drive the pandemic and where they struggle to cope. The current published information largely focuses on the nature of the disease and the potential effect on people especially in terms of mental health, social distancing, and being quarantined [15,16,17,18]. The methodology mainly focuses on reviews and there are a few primary research studies. However, to the researchers’ knowledge, there is no known research regarding the lessons learnt by South African families from the COVID-19 lockdown experience. This information does not exist, but it is important to know how families function and cope during this difficult time so that it becomes a resource of coping and resilience.

At a period like this, it can be difficult to establish any positives. However, history has shown that major catastrophes can serve as valuable catalysts for societal change [19,20]. Besides providing possible guidelines for how future pandemics can be handled, COVID-19 has also provided valuable insights on how to improve family dynamics and well-being. Thus, while the world endures to fight the pandemic and come to grips with the “new normal”, it is important to reflect on the valuable life lessons the pandemic has had on families and keep sight of these while striving to build on what has been learnt. Thus, the main purpose of this paper is to explore the perceptions of South African families on lessons learnt during the COVID-19 lockdown.

## 2. Materials and Methods

### 2.1. Participants

A total of 31 members from different families participated in the study. This total sample was determined by data saturation, a point when the research assistants decided to stop interviewing further participants due to information redundancy. The family members voluntarily participated through a process of snowball sampling. The research assistants initially approached families living in their vicinity and those family members whose contact details they had, requesting their participation. These participants then referred the authors to other families for participation in the study. This process developed into a snowball non-probability sampling technique. The inclusion criteria for this study were that the family members had to be above the age of 18 years and residing in one of the six municipal districts of the Western Cape Province (Overberg District Municipality, Central Karoo District Municipality, Winelands District Municipality, City of Cape Town Metropolitan Municipality, Eden District Municipality, West Coast District Municipality). Participants were excluded if they were not familiar with one of three languages (i.e., English, Afrikaans, and Xhosa) that were the predominant languages in the selected areas. The demographic characteristics of the participants are summarised in Table 1.

### 2.2. Procedure

All study procedures were approved by the Human Social Sciences Research Ethics Committee of the University of the Western Cape. Due to the strict COVID-19 lockdown restrictions, which were in place at the time of the study, participants were duly informed telephonically of the topic under investigation, the benefits of the study, and the manner and form in which data were collected and how confidentiality was upheld. Thereafter, participants were provided with the opportunity to ask questions and were requested to provide verbal consent if they were willing to participate before the interviews were initiated. Once the informed consent was given, a telephone interview was scheduled at the participants’ convenience. A few participants chose not to partake in the interviews after receiving the full study information mainly due to a lack of time.

Research assistants were employed to help in screening the participants for eligibility and collecting the data through semi-structured qualitative interviews. The semi-structured interviews (see Table 2) comprised a series of open-ended questions to help in the understanding of participants’ derived lessons from the COVID-19 lockdown experience. The interviews also included probe questions for the participants who struggled to convey their views to the interviewers. The interviews took between 30 and 60 min and were audio-recorded. All interviews were held between June–September 2020.

### 2.3. Data Analysis

The digital recordings were transcribed verbatim using a professional transcribing service and the data were then analysed inductively using thematic analysis [21]. As the interviews were also conducted in Afrikaans and Xhosa, the transcripts were carefully translated into English from the original language before the analysis procedure was performed. The analysis process (see Table 3) firstly involved the initialisation phase which involved the researchers familiarising themselves with the data through reading and re-reading the transcripts in order to derive meaning then after they produced preliminary codes. Coding reduces the amount of raw data to what is relevant to the research question, breaks the data down into manageable sections, and takes researchers through the transformation of raw data [21]. The last stage of the initialisation phase involved writing reflective notes a process, which allows researchers to remain faithful to participants’ perspectives and improve the validity of theme development. After the completion of coding, themes generation was commenced by categorising different codes into related clusters. The themes were then termed and well-defined in order to derive sense from the main ideas developing from them. Thereafter, all the defined themes were checked against the original transcript to ensure that the text confirmed the classification. Subsequently, the final analysis and writing up of findings were carried out. This final phase involved a holistic written commentary describing and connecting various themes to quotations by the participants. The finalisation phase also strived to address the study main aim.

### 2.4. Trustworthiness and Rigour of the Study

The credibility, transferability, conformability, dependability, and reflexive approaches to the inquest and analysis were used to establish the rigour and trustworthiness in this study. A comprehensive methodology—the site of study, participants and data collection procedures—was provided to ensure transferability in this study. The methods of analysis were also described in detail to ensure dependability for the study. Furthermore, a single interview schedule developed by all members of the research team was used for all interviews. The data coding was conducted by more than one member of the research team. The credibility of the study was ensured at the end of each interview by conducting member checking as well as conducting a recap of the main points from the interviews. In addition, participants were allowed to express themselves freely during the interview. To ensure the confirmability of the study, verbatim transcripts of the participants’ responses were provided. The research team kept a reflective journal containing the discussions, deliberations, and decisions made as a part of the audit trail when conducting this study.

## 3. Results

One of the main themes identified in the broader study on COVID-19 and family life as identified by the respondents pertains to the lessons learnt as a result of the COVID-19 pandemic, particularly during the hard lockdown period. These lessons are categorised according to the sub-themes (see Figure 2) from the data which are as follows: (1) Hygiene and health consciousness: concerned with a higher expectation of overall cleanliness and an awareness of the healthiness of one’s diet and lifestyle respectively; (2) Appreciation for family: entails the expression of gratitude towards family members; (3) Valuing life: involves the state of cherishing every moment of one’s life and making the best out of everything; (4) Self-Introspection: encompasses the examination of one’s own conscious thoughts and feelings; (5) Less dependency: operationalised in the study as the act of relying less on friends, family and colleagues in partaking on individual’s daily activities; (6) Remote working; refers to work executed outside of a classic office environment, also regarded as working from home and (7) Financial savings: involves the preservation of money and a reduction in avoidable expenditures.

The presentation of each sub-theme reinforced by the quotes as expressed by the participants is outlined below.

### 3.1. Hygiene and Health Consciousness

One prominent sub-theme from the responses that was identified as a key lesson by the participants was around the issue of hygiene and health during the COVID-19 lockdown. With regards to hygiene, participants indicated that due to the COVID-19 pandemic there was now a higher expectation of overall cleanliness among families than usual. The extracts shown below illustrate this:


*“People are generally just not hygienic, uhmm and then you realise how clean certain places are now. People have actually had to clean their tables, clean trolleys when you go to the shops and everything just looks and feels cleaner… so I think the level of hygiene has improved throughout the community” (Male, 26 years old, human resource management associate).*



*“As I was always emphasising, it is more to do with cleanliness and yeah, just to ensure personal hygiene” (Male, 30 years old, PhD graduate).*


Good hygienic practices have been proven to reduce harmful, disease-causing bacteria, germs, and viruses. The above responses by the participants indicate that, resulting from the pandemic, participants were more aware of the benefits of maintaining clean environments in order to reduce the transmission of COVID-19, and lessen health risks for themselves, their loved ones, and communities. In addition to the remarkable improvement in general cleanliness, other respondents were more specific on how they learnt to improve their personal hygiene due to the COVID-19 pandemic through regular hand washing and sanitisation, as well as wearing face masks. This is evident by the following statements by the participants:


*“Two things I’ve learnt. It’s a tough one. Oh, washing your hands and sanitising is very important” (Female, 21 years old, student).*



*“I think there is just so much more things that you think about that you would not have thought about in the past. like, okay, do I have some sanitisers?” (Female, 35 years old, attorney).*



*“I have learnt to protect others because when you put on your mask or whatever, you are not just doing it for yourself” (Female, 26 years old, Graduate Intern).*


Besides COVID-19 having resulted in families becoming increasingly hygienic, some participants revealed that they also learnt to take an active interest in their health. Particularly, they stressed being more conscious of what they consumed and the risks of getting infected.


*“COVID has taught us to watch what we eat and what we drink and also to cover yourself when you are not feeling well” (Female, 70 years old, pensioner).*



*“Eating healthy…A healthy lifestyle is important” (Female, 21 years old, student).*



*“I think to be more aware of a person’s health and well-being” (Female, 64 years old, Administrator).*


From these statements, it can be noted that the participants were more mindful of the food they consumed, mainly the participants perceived eating healthy as being imperative in order to boost their immune system for fighting the virus and reduce the risk of infections. Furthermore, the statement by the 64-year-old, female administrator highlights the importance of constantly checking on an individuals’ health and well-being in preparation to fight future pandemics such as the COVID-19. Lastly, an important issue that also emerged from most participants’ responses concerns safeguarding one’s health against the outbreak of viruses going onwards specifically by wearing of face masks, physical distancing, sanitising and regular hand washing.

### 3.2. Appreciation for Family

Some Participants also identified the issue of appreciation for the family during the COVID-19 pandemic as a key lesson learnt during the COVID-19 lockdown period. The participants revealed that the pandemic presented them with the opportunity to understand their families by dedicating more time for family, either physically or virtually, than ever before. While for others, the pandemic taught them to care more for family members as the lockdown restricted them from visiting family at all. Some of the participants relayed their opinions as follows:


*“Yes, I have learnt a lesson on how much you care for your family when you are not with them and through the distance from your family you realise how much you mean to each other. We are not getting functions together, we are not enjoying each other’s company and going out together” (Female, 64 years old, pensioner).*



*“My idea of family may have changed a lot, through it all, I have learnt that the pandemic has been helpful in reminding ourselves of the family members in our lives who are there for us, whether in person or over the phone” (Male, 39 years old, General Manager).*



*“For the family, we learnt to engage with each other, we learnt to have more unity than before” (26 years old, female, graduate Intern).*



*“I realised that no matter how much time we think we have; at the end of the day, what I came to appreciate was that we simply don’t spend enough quality time with our families” (Male, 27 years old, Social worker).*


The above extracts from the participants show that, amid the growing pandemic fears and restrictions, appreciation for family increased. For those who managed to engage with their families during the hard lockdown period, it enhanced the strengthening of the family unity as evident by the response, for instance from a 26 years old, female graduate Intern. In addition, it also exposed how during pre-COVID-19 families were not spending adequate time together. On the other hand, for those who could not be with their families, the feeling of being isolated made them realise how much they cared for their families.

### 3.3. Valuing Life

Another sub-theme that was derived from the data and emerged as a key lesson from the COVID-19 lockdown experience for the participants was that of valuing life. One of those issues as revealed by the participants which they considered an important lesson is to value one’s life and that of others. This is aptly illustrated by the responses below:


*“lessons wise I would say value the time that you have because you never know” (Male, 27 years old, counsellor).*



*“COVID has taught us to appreciate our lives more” (Female, 70 years old, pensioner).*



*“…. just make the most of every opportunity that comes your way because you never know what is going to happen” (Female, 56 years old, tech business partner).*


In emphasising the importance of valuing life one participant even went on to mention how everyone, irrespective of income status or fame needs not to take life for granted. She had this to say:


*“Life is very fragile. Money and fame means very little in the grand scheme of things” (Female, 26 years old, law student).*


The quotations described above indicate that life is unpredictable, therefore the participants are advocating that people should appreciate being alive and make the best out of the time they have in life.

### 3.4. Self-Introspection

The COVID-19 pandemic also presented a rare opportunity for some family members to understand themselves better through self-introspection. knowing yourself better results in a clearer sense of purpose, greater well-being, self-acceptance, and happiness [22]. The extracts shown below illustrate the participants’ view regarding self-introspection during the COVID-19 lockdown:


*“Self-reflection, just more time to yourself and for some it was like a reflection of where do I stand” (Female, 21 years old, student).*



*“I would say, the first lesson was, I think, I noticed that most people I could see were reflecting on the work they were doing they were asking themselves, Is this actually something I want to do?….Do I actually want to be here and do this?” (28 years old, Software engineer).*



*“Because we were dealing with a lot of things. We needed the world to like keep quiet so that you can find yourself” (Female, 26 years old, graduate Intern).*


Due to the COVID-19 lockdown, most people had adequate time in their homes away from work and social activities. This as described by the participants gave them an opportunity to self-introspect and think deeply of what was important in their lives. In addition, the effects of the lockdown also gave other people an opportunity to discover who they really are and their goals in life.

### 3.5. Less Dependency

As humans, we are naturally “wired” to connect with friends, family, and colleagues with most of the daily routines being influenced by how we associate with each other. However, through the enforcement of the initial hard lockdown measures, especially the restrictions on movement, life took an unexpected turn as the pace of life drastically slowed down, and many people felt isolated. The period hence required some changes in behaviour and one of the lessons learnt from leaning into the unnerving feeling of solitude as indicated by some of the participants was that of becoming less dependent. This is highlighted by the responses below.


*“Uhmm, how to be more independent. Yeah, not just like you use to. Just a random example, I would always go run with friends and now we cannot run with friends” (Female, 28 years old, research assistant).*



*“So, I’d say also just being okay to saying ‘no’ to going out with friends and stuff from time to time should be acceptable and people should do it a little more often. Not a lot more often, but a little more often” (Male, 28 years old, Software engineer).*



*“For me I’ve learnt that actually I can do a lot alone. That also includes the life about having fun, life without relationships. That’s one of the lessons that I learnt” (Male, 23 years old, student).*



*“When the lockdown was initially announced, I became hesitant on how I was going to manage all alone. I am however surprised that It is not all about being surrounded by friends for me to be happy as I am coping so well” (Male, 30 years old, graduate).*


The above-outlined quotations show that resulting from the COVID-19 lockdown, people became more resilient and sought some coping mechanisms to survive away from their families and friends. By so doing, people learnt that they have capabilities to cope even while in isolation.

### 3.6. Remote Working

As the COVID-19 movement restrictions disrupted work activities, respondents revealed that they learnt how to adapt to the situation specifically through working remotely. With advancements in information and communication technologies (ICTs), and especially with the greater accessibility of the internet, remote working (also referred to as work from home) has grown in its use as a common mode of work in South Africa and globally from the onset of the pandemic. Some participants had the following to say with regards to working from home:


*“I can still work and make a valuable contribution from my bed. Yeah. So, we can do things differently and still do things well in this country” (Female, 46 years old, monitoring and evaluation officer).*



*“I feel allowing people the opportunity to work from home has been very beneficial” (Male, 26 years old, ESL teacher-Currently unemployed).*



*“I have learnt on how to work from home effectively” (Female, 28 years old, research assistant).*


The above statements imply that people had to learn new innovative ways of working as a result of the COVID-19 lockdown restrictions and some even realised that they could be much more productive whilst working remotely.

### 3.7. Financial Savings

The COVID-19 pandemic caused a big disruption in the lives of most South Africans, with those living in low-income communities being affected the most. The lockdown caught many off-guard and left people with little to get by. Even with the help of the government, many South Africans are still failing to support their families. One important lesson that the COVID-19 lockdown experience taught people, as highlighted by the respondents, is that it pays to save financially. This is evident by the following statements by the participants:


*“It has actually been sort of in a way an eye-opener for people to relook at their finances also to make sure that they have got money set aside. So, in the event something like this happens they have got money to rely on to sort of help them with their everyday stuff” (Female, 41 years old, private banker).*



*“I have learnt that you must save money for future” (Female, 26 years old, graduate Intern).*



*“In times like this when you know people are losing jobs and everything and if you have some savings it will really help you and carry yourself through” (Male, 31 years old, educator).*



*“Uhmm, I think one would be, it does not necessarily apply to me, but it is one that applies to many people. People need to plan for the future more because there were many that were not prepared and did not have the financial ability to pay for things that they needed during the lockdown” (Male, 26 years old, ESL teacher- Currently unemployed).*


Prior to COVID-19 most families had a guarantee that they would receive some income monthly, but due to COVID-19, which led to the closure of most economic activities, some breadwinners lost their jobs, while some got their salaries reduced. During this period people who had savings managed to keep going but those who had no savings were in difficult positions. This was highlighted by the responses taught many people the importance of saving financially as a cushion against unforeseen circumstances.

## 4. Discussion

The aim of the study was to provide an understanding of the lessons learnt by South African families during the COVID-19 lockdown. Findings from the study are summarised into seven sub-themes: hygiene and health consciousness, appreciation for family, valuing life, self-Introspection, less dependency, remote working, and financial savings. These findings reveal novel insights into how specifically the initial stages of the COVID-19 social restriction/isolation measures offered lessons to South African families.

Based on the shared views from the respondents who participated in the study, there is evidence of greater caution related to health and hygiene among South African families than usual due to the COVID-19 pandemic. This signifies a very important lesson emanating from the lockdown experience. In support of this finding, another study reported that the COVID-19 pandemic has, globally, taught people the body’s response to emerging diseases and through it all, the importance of basic hygiene [23]. This assertion proves that the ongoing COVID-19 pandemic has unearthed the paramount significance of the practical application of basic concepts of public health, which were now fairly considered trivial, such as healthy eating, personal hygiene, personal protective equipment, or basic epidemiological measures [24]. As the COVID-19 crisis continues with new strains emerging, people are therefore consistently adjusting their hygiene and health strategies for dealing with the pandemic.

Appreciation for family was also noted in this study as a key lesson emerging from the COVID-19 lockdown experience. The participants highlighted how the pandemic has been instrumental in reminding themselves of the people who have been there for them in their lives, whether in person or over the phone. For some, the pandemic meant spending more time with family than ever, while for others, it meant not seeing family at all. In both cases, despite being different, the participants were of the same view that the pandemic taught them to have a greater appreciation for family. The respondents’ views are not peculiar, as studies conducted during the same period in other countries revealed similar findings. For instance, a study on the psychological well-being during COVID-19 lockdown [25], which was conducted in The Kingdom of Saudi Arabia, revealed that a large proportion of the participants felt that their appreciation for family got stronger during the lockdown. In addition, a study that further focused on British Families in Lockdown established that families had positive experiences during the lockdown, particularly in terms of strengthening family bonds [26].

By spending much of their lockdown time with minimum influence from peers, some respondents revealed having learnt to improve their self-awareness through self-introspection. This indicates that, despite the COVID-19 lockdown being a difficult moment, it presented participants with the opportunity to discover themselves. The development of self-introspection appears to facilitate healing from difficult situations [27] and to find constructive solutions to calamities. Respondents also reported that the COVID-19 pandemic had changed the manner in which they viewed life. Particularly, the possible presence of imminent death of so many globally because of the COVID-19 pandemic for instance has been devastating and a key lesson that has caused a reassessment of the value of human life. The impact of COVID-19 is increasingly drawing increased attention toward the need to value lives due to severe illness and the premature deaths being witnessed across the globe [28]. The participants emphasised the importance of making the most of the present moment as the pandemic has proved how life is not guaranteed.

The results from the study also reflect on how the respondents learnt about the importance of saving money from the onset of the COVID-19 lockdown restrictions. Almost half of the households experienced financial challenges in South Africa over the lockdown period, such as a struggle to pay debts or a reduction in household income [29]. For households that were vulnerable financially before the COVID-19 pandemic, their situation has deteriorated, particularly those from low-income communities. As such, several participants noted how they and their families need to relook at their finances, particularly through saving for unforeseen circumstances. Furthermore, evidence shows that, as the world grapples with the aftermath of COVID-19, many South Africans have started saving for emergencies in case things deteriorated through measures such as drawing monthly budgets, stokvels, setting up automatic savings contributions, and cutting back on subscriptions [30].

Another important finding from the study as outlined by most participants relates to remote working. Participants revealed that, since the inception of the pandemic, they have learnt that they could be just as productive as they were at work premises through remote working. This implies that South Africans have a transformed shift of interest towards the desire to work remotely even post lockdown. The finding is consistent with an earlier report that indicated 40% of South African professionals articulating their desire to move to remote working on a full-time basis, with a further 27% preferring at least 50% remote working [31]. The report further anticipated that some of the variations merged into workplaces because of the COVID-19 pandemic will be more enshrined in daily working settings going forward, and for some sectors, there will be a component of remote working entrenched for good. People who work from home are able to strike a manageable work-life balance and have more time to spend with family which improve mental health, and ultimately trigger a happier more productive workday [32]. However, this is still an issue facing heated debate as some studies have not found clear effects of remote working on job productivity [33,34].

The results of this study have implications for policy and practice in developing nations including South Africa. For instance, the results of the study should prompt policymakers to develop appropriate policies that will encourage families to maintain healthy and hygienic habits as this will ease the burden on families during future pandemics as proper health and hygiene are important for preventing both the spread of infections and loss of loved ones. It is also crucial for governments in developing nations to come up with community programmes that foster a culture of financial savings as this will provide a cushion against financial shocks on families particularly in times of crisis. Furthermore, since many families are now working from home (the use of the internet has increased), this should prompt governments in developing nations, including South Africa to bridge the digital divide present in most of these countries. The internet has been a vital source of strengthening family ties as well during the lockdown as it afforded family members an opportunity to interact with each other irrespective of geographic boundaries. Video calling, for example, gained prominence in Africa as an important way of communication during the hard lockdown period. Platforms such as Google Meets, Zoom, WhatsApp video calling allowed family members to interact in a more natural way by enabling face-to-face communication in real-time. The internet has reduced distances and brought families closer hence developing nations need to invest adequately in ICT infrastructure.

The major strengths of the present study encompassed the qualitative attention on understanding the lessons learnt by South African families during the COVID-19 lockdown. Due to the unanticipated nature of the pandemic, the development of a deeper understanding of the phenomenon without the limits of traditional quantitative measures is required. In the current study, this led to a further understanding of the themes specifically related to the positive insights resulting from the effects of the lockdown measures on South African families. The study should also be appraised considering its limitations. Initially, the interviews were scheduled to be conducted face-to-face, however, due to COVID-19 movement restrictions they were conducted online via Google Meets or Zoom and telephonically. This may have limited our in-depth understanding of families’ experiences as the participants’ non-verbal expressions could not be observed. It is also significant to highlight that the applied sampling method (Snowballing) might have prevented the selection of a true representative of the study population. Finally, there is a need to further conduct a follow-up study post the pandemic to assess whether the lessons identified in the study will be retained by most families using the same setting. Future studies are also needed to be conducted to evaluate the effectiveness of remote working in terms of balancing family time and work commitments and productivity as this is still a contentious issue. Last, there is a need for similar future studies to employ a mixed-method approach in order to establish relationships emerging from the data such as the association between the participant characteristics and factors such as infection influence, as well as government containment measures.

## 5. Conclusions

As evidence has emerged regarding COVID-19 lockdown experiences for South African families, it has become clear that despite causing hardship for many, the pandemic has provided valuable lessons for families to examine the way they have been living and strive to improve family life for the better. One of the two common lessons from the COVID-19 lockdown experience for families has been an increase in hygiene and health consciousness as no longer can societies function without integrating health values into their practices under the new normal. Besides this, the other significant consequence of the lockdown has been that of greater appreciation for family, which fosters closeness and unity. Other lessons also emanating from the study that benefit family members include valuing life, an opportunity to improve self-awareness through self-introspection, the ability to be less dependent, working productively from home, and the benefits of financial savings. The study strongly recommends the realisation of such lessons by South African families even post the pandemic. The lessons emerging from the study has the potential of strengthening families to be a resource of coping and resilience during very difficult times at the same time, contributing to greater physical, social, and economic functioning of families across South Africa. There is, therefore, a need for family policies to take into consideration the development of strategies that taps into the strengths of families in order to enhance family well-being in times of future pandemics.

## Figures and Tables

**Figure 1 ijerph-18-12552-f001:**
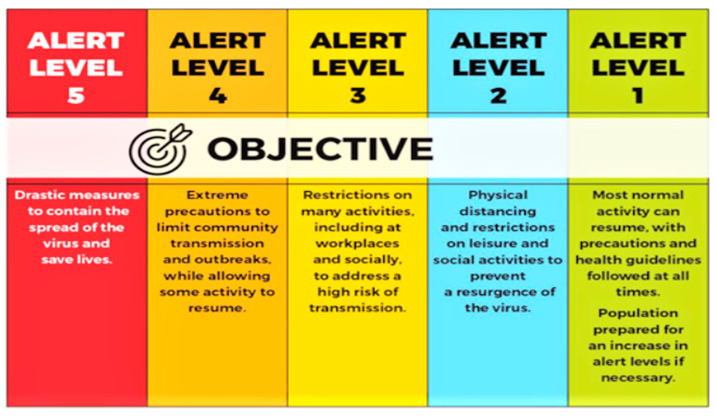
Summary of alert levels.

**Figure 2 ijerph-18-12552-f002:**
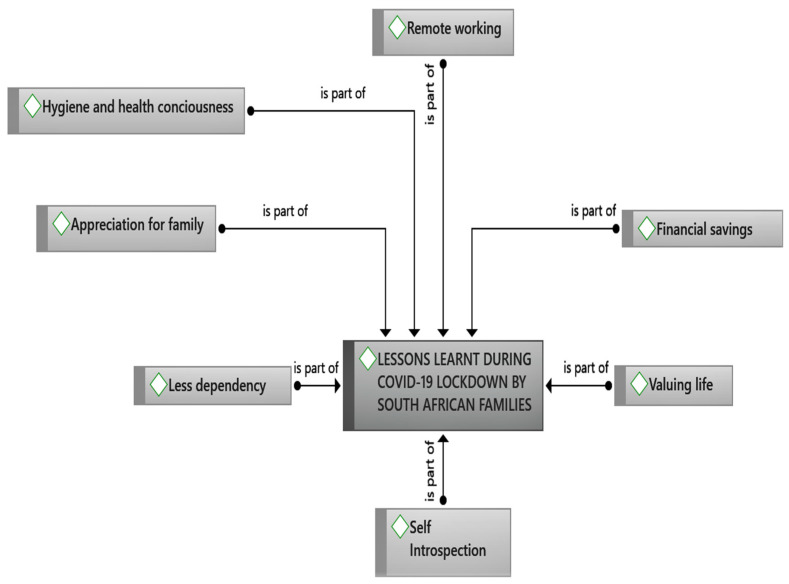
Summary of the study sub-themes.

**Table 1 ijerph-18-12552-t001:** Demographic characteristics of the participants.

Demographic Characteristic		Frequency
Gender	Male	13
	Female	18
Age	18–24 years	4
	25−35 years	18
	36−46 years	3
	47+ years	6
Employment status	Employed	21
	Unemployed	10
Family structure	Nuclear family	22
	Single parent family	2
	Grandparent family	2
	Extended family	5
Total		31

**Table 2 ijerph-18-12552-t002:** Semi-structured interview guide.

Questions
Tell me about the experiences of your family during the COVID-19 lockdown? What are the main concerns you have for this virus? How do you think families in general are coping with COVID-19 during this period? Could you describe any positives of the COVID-19 lockdown experience as a family? What aspects of the lockdown do you wish to be retained post the pandemic? What are the major challenges the family faces right now and how are they handled? What have you derived so far from the COVID-19 lockdown experience?

**Table 3 ijerph-18-12552-t003:** Summary of phases and stages of theme development used in the study.

Phases	Stages
Initialisation	Reading and re-reading transcriptions and establishing meaning; Coding; Reflective notes
Generation	Categorising; Comparison; Terming and defining
Verification	Checking themes against original transcripts
Finalisation	Final analysis and writing up of findings

## Data Availability

The anonymised transcribed interviews are available from the corresponding author upon reasonable request.

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
