# Peer review of "Lessons Learnt during COVID-19 Lockdown: A Qualitative Study of South African Families"

_ijerph, 2021, doi:10.3390/ijerph182312552_

Round 1
Reviewer 1 Report
It was a pleasure reviewing the manuscript titled, “Lessons learnt during COVID-19 lockdown: A qualitative study of South African families”. Overall, the manuscript is expertly reported with the study main aim and question clearly stated which makes it easier for the reader to follow.
The research approach that was used to guide the data collection and analysis was also aptly explained in the manuscript. This includes the design used, sampling method, data collection tools and procedure as well as the data analysis process.
Furthermore, the findings of the study were appropriately presented and discussed in relation to the existing literature. I would therefore wish to recommend that the manuscript be considered for publication after the authors address the following issues;
- On the introduction section, there is also need to provide the global COVID -9 statistics within the global context.
- The manuscript has to be thoroughly re-read and corrected for some slight grammatical mistakes as some words / letters are missing.
- Despite the study recommending for realisation of the established lessons from the lockdown experience by South African families even post the pandemic, there is a need for the article to also propose recommendations for policy.
- Please check the list of references for consistence as the prescription of the journal should be followed throughout. For instance, for online sources, the date accessed appears as follows; (Accessed on 29 September 2021) or in some instances with a colon after ‘on’ as follows; (Accessed on: 29 September 2021).
Author Response
Dear Reviewer 1
We are grateful for the extensive feedback which we received from you of our manuscript. We have attended to the issues you raised and hereby present our point by point feedback. We are grateful for the insightful feedback from you as their remarks helped us improve the quality of our manuscript. We hope that our revised manuscript will be found appropriate for publication. Kindly find below, our responses (in bold and italics) to the reviewers’ feedback (plain text).
Thank you,
Sincerely,
Gift Donga (corresponding author).
Authors response:
We appreciate the reviewer’s input and have incorporated the following changes into the manuscript:
- On the introduction section, there is also need to provide the global COVID -19 statistics within the global context
- Global stats provided check first paragraph line 6 on page 1
- The The manuscript has to be thoroughly re-read and corrected for some slight grammatical mistakes as some words / letters are missing
- We appreciate the reviewer’s input and have re-read the manuscript and attended to grammatical errors in text.
- Despite the study recommending for realisation of the established lessons from the lockdown experience by South African families even post the pandemic, there is a need for the article to also propose recommendations for policy.
- Recommendation for policy provided in paragraph 7 under discussion on page 11
- Please check the list of references for consistence as the prescription of the journal should be followed throughout. For instance, for online sources, the date accessed appears as follows; (Accessed on 29 September 2021) or in some instances with a colon after ‘on’ as follows; (Accessed on: 29 September 2021).
- We appreciate the reviewer’s input and have corrected the reference list as suggested

Reviewer 2 Report
This study is a qualitative study to interview 31 participants in South Africa about the lesson from Covid-19 pandemic. The information about the influence of Covid-19 pandemic in South Africa to the general population is very important, however, the author should write some national statistic data including Covid-19 pandemic, etc.
- Even this study is a qualitative study, the author should make presentation scientific sound, such as descriptive analysis in Table 1.
- It should be write study purpose in scientific sentence, not question..?
- The data analysis is not appropriate, it should include some summary of table in descriptive way.
- It is unclear what is the relationship between participant characteristics and its data analysis results. For example, the relationship between economic status and infection influence, etc.
- It should be mentioned about the recent history of lockdown and exact data in city or national level infection number, etc.
- The authors should consult public health expert about this study, and should be advised the way of presentation.
- Please discuss and provide message for the developing nations regarding this study topic.
Author Response
Dear Reviewer 2
We are grateful for the extensive feedback which we received from you of our manuscript. We have attended to the issues you raised and hereby present our point by point feedback. We are grateful for the insightful feedback from you as their remarks helped us improve the quality of our manuscript. We hope that our revised manuscript will be found appropriate for publication. Kindly find below, our responses (in bold and italics) to the reviewers’ feedback (plain text).
Thank you,
Sincerely,
Gift Donga (corresponding author).
Authors response:
We appreciate the reviewer’s input and have incorporated the following changes into the manuscript:
Even this study is a qualitative study, the author should make presentation scientific sound, such as descriptive analysis in Table 1
We appreciate the reviewer’s input and we added some tables and figures (see Table 2 and 3; as well as Figure 1 and 2)
- It should be write study purpose in scientific sentence, not question..?
Authors response:
This has been rectified. Check the last sentence on the Introduction section
- The data analysis is not appropriate, it should include some summary of table in descriptive way.
Authors response:
A summary of the qualitative analysis provided in Table 3
- It is unclear what is the relationship between participant characteristics and its data analysis results. For example, the relationship between economic status and infection influence, etc.
Authors response:
Again we appreciate the reviewers feedback, however with the available data we gathered it was impossible to report of any relationships. Thus due to the significance of the comment by the reviewer we suggested that; there is need for similar future studies to employ a mixed method approach in order establish relationships emerging from the data such as the association between the participant characteristics and factors such as infection influence as well as government policies, for instance the lockdown restrictive measures. (check last sentence on the discussion section)
.
- It should be mentioned about the recent history of lockdown and exact data in city or national level infection number, etc.
Authors response:
The recent history of lockdown has been included ( check paragraph 3 of the introduction section).
- The authors should consult public health expert about this study, and should be advised the way of presentation.
Consulted with one expert from the University of the western Cape who offered his suggestions. We also checked past qualitative studies (e.g., https://www.mdpi.com/1660-4601/18/5/2768 ; https://www.mdpi.com/1660-4601/17/8/2953 ; https://www.mdpi.com/1660-4601/17/8/2826 ) published by mdpi- Int. J. Environ. Res. Public Health to get some guidelines.
- Please discuss and provide message for the developing nations regarding this study topic.
Message for developing nations included (Check paragraph 7) under the discussion section

Reviewer 3 Report
This is an interesting study that explores the lessons learnt during 12 COVID 19 by South African families.
I think the authors can improve on the reporting, particularly the methods section. For example, although snowball sampling was used, it is not stated how the first set of respondents were selected and what informed the halt of the snowballing process (saturation?).
Was the analysis inductive or deductive?
The authors should endeavor to use the COnsolidated criteria for REporting Qualitative research (COREQ) checklist as a guide for reporting (http://cdn.elsevier.com/promis_misc/ISSM_COREQ_Checklist.pdf)
Author Response
Dear Reviewer 3
We are grateful for the extensive feedback which we received from you of our manuscript. We have attended to the issues you raised and hereby present our point by point feedback. We are grateful for the insightful feedback from you as their remarks helped us improve the quality of our manuscript. We hope that our revised manuscript will be found appropriate for publication. Kindly find below, our responses (in bold and italics) to the reviewers’ feedback (plain text).
Thank you,
Sincerely,
Gift Donga (corresponding author).
Authors’ response:
I think the authors can improve on the reporting, particularly the methods section. For example, although snowball sampling was used, it is not stated how the first set of respondents were selected and what informed the halt of the snowballing process (saturation?).
- The issue of how the first set of respondents were selected has been clarified. Check section 2.1 page 3 line 3
- The issue of saturation has been clarified in section 2.1 paragraph 1 line 2
Was the analysis inductive or deductive?
- The analysis was inductive this has now been mentioned in section 2.3 line 2.
The authors should endeavor to use the COnsolidated criteria for REporting Qualitative research (COREQ) checklist as a guide for reporting (http://cdn.elsevier.com/promis_misc/ISSM_COREQ_Checklist.pdf)
- We appreciate the reviewer’s input and we have used the suggested checklist which has helped immensely in improving the manuscript.

Round 2
Reviewer 2 Report
I confirmed the improvement of quality of study.